# Genotyping *Echinococcus*
*multilocularis* in Human Alveolar Echinococcosis Patients: An EmsB Microsatellite Analysis

**DOI:** 10.3390/pathogens9040282

**Published:** 2020-04-13

**Authors:** Jenny Knapp, Bruno Gottstein, Stéphane Bretagne, Jean-Mathieu Bart, Gérald Umhang, Carine Richou, Solange Bresson-Hadni, Laurence Millon

**Affiliations:** 1UMR CNRS 6249 Laboratoire Chrono-environnement, Université Franche-Comté, 16 Route de Gray, 25030 Besançon, France; dr.bresson.hadni@wanadoo.fr (S.B.-H.); laurence.millon@chu-besancon.fr (L.M.); 2Department of Parasitology-Mycology, National Reference Centre for Echinococcoses, University Hospital of Besançon, 25030 Besançon, France; 3Institute for Infectious Diseases, Faculty of Medicine, University of Berne, 3001 Berne, Switzerland; bruno.gottstein@vetsuisse.unibe.ch; 4Parasitology-Mycology Laboratory, Lariboisière-Saint Louis-Fernand Widal Hospital, Assistance Publique-Hôpitaux de Paris (AP-HP), Université de Paris, 75475 Paris, France; stephane.bretagne@aphp.fr; 5UMR INTERTRYP, IRD/CIRAD, University of Montpellier, 34398 Montpellier, France; jean-mathieu.bart@ird.fr; 6ANSES, Nancy Laboratory for Rabies and Wildlife, Wildlife Surveillance and Eco-Epidemiology Unit, Technopôle Agricole et Vétérinaire, B.P. 40009, 54220 Malzéville, France; Gerald.UMHANG@anses.fr; 7Department of Hepatology, University Hospital of Besançon, 25000 Besançon, France; crichou@chu-besancon.fr

**Keywords:** Alveolar echinococcosis, *Echinococcus multilocularis*, European endemic area, contamination event, genotyping, microsatellite EmsB

## Abstract

For clinical epidemiology specialists, connecting the genetic diversity of *Echinococcus*
*multilocularis* to sources of infection or particular sites has become somewhat of a holy grail. It is very difficult to trace the infection history of alveolar echinococcosis (AE) patients as there may be an incubation period of five to 15 years before reliable diagnosis. Moreover, the variability of parasitic manifestations in human patients raises the possibility of genetically different isolates of *E. multilocularis* having different levels of pathogenicity. Thus, the exposure of human patients to different strains or genotypes circulating in geographically different environments may lead to different disease outcomes. Molecular tools, such as the microsatellite marker EmsB, were required to investigate these aspects. This genetic marker was previously tested on a collection of 1211 European field samples predominantly of animal origin, referenced on a publicly available database. In this study, we investigated a panel of 66 metacestode samples (between 1981 and 2019) recovered surgically from 63 patients diagnosed with alveolar echinococcosis originating from four European countries (France, Switzerland, Germany, Belgium). In this study, we identified nine EmsB profiles, five of which were found in patients located in the same areas of France and Switzerland. One profile was detected on both sides of the French-Swiss border, whereas most patients from non-endemic regions clustered together in another profile. EmsB profiles appeared to remain stable over time because similar profiles were detected in patients who underwent surgery recently and patients who underwent surgery some time ago. This study sheds light on possible pathways of contamination in humans, including proximity contamination in some cases, and the dominant contamination profiles in Europe, particularly for extrahepatic lesions.

## 1. Introduction

*Echinococcus multilocularis* is the parasite responsible for alveolar echinococcosis (AE) in humans, one of the most dangerous zoonoses in the Northern Hemisphere. It belongs to the Taeniidae family and its life cycle involves passage through several different mammalian hosts. Canine carnivores, particularly red foxes in Europe, are the principal definitive host (DH) and they harbor these worms in their intestines. Experimental estimates of the patent period in carnivores range from 25 days post protoscolex ingestion [1] to beyond 90 days [2]. Rodents act as an intermediate host (IH), following the ingestion of parasite eggs originating from infected carnivore feces contaminating the environment. The oncospheres hatched from the eggs reach the liver of the IH where they develop into metacestodes, and protoscoleces are usually produced two to three months post-infection. Infection ends in the death of the rodent IH within five to eight months of infection. Humans are generally considered to be aberrant IH and protoscoleces rarely develop in infected humans. An estimated 18,000 new cases of AE per year occur worldwide [3], with 1600 cases in Europe and 16,400 in China. It is often challenging to identify the temporal and spatial circumstances surrounding human infection. Indeed, the first symptoms occur after an initial asymptomatic incubation period of 5–15 years [4], and the multiple risk factors [5,6,7] make it almost impossible to trace transmission. Living in an endemic region is the major risk factor for contracting the disease [7], but many other putative risk factors have been identified, including agricultural activities, hunting activities and owning a pet dog or cat [8]. 

In recent decades, AE human cases have been reported in large parts of Central and Northern Europe, from Eastern Romania [9] to Western France (data from the National Reference Centre for Echinococcosis (NRC-E), Besançon, France). Moreover, *E. multilocularis* parasites have been found in foxes from Central Romania to Western France [10,11], so Europe as a whole may be considered as a large endemic area. Furthermore, the French patients were diagnosed outside the regions in which this parasite is generally considered endemic (e.g., Brittany and the Pyrenees) ([7], (data from NRC-E), raising questions about the precise limits of the current endemic area. Information about the place of residence of the patients before diagnosis can be obtained from national alveolar echinococcosis registries, such as the NRC-E in France, but the geographic site and timing of infection remain open to speculation. The big question is whether it is possible to associate the strains infecting patients with strains from foxes, given the presumed long interval between infection and diagnosis. This would require a given *E. multilocularis* strain to remain in a given geographic environment for almost 10–15 years. Another epidemiological factor identified in a recent study is the opportunistic nature of infections with this parasite in patients with immunosuppression [12]. Does the parasite strike human hosts indiscriminately, or are some strains more likely to infect humans than others? Genetic tracking methods could provide important elements for improving our understanding of these parasite infection events. Nuclear and mitochondrial genes are generally used to assess the genetic diversity of the parasite at continental level [13,14,15], but the resolution of these markers is too low for analyses at the local scale. Microsatellite markers are highly polymorphic tools that have been used to describe genotypes in *E. multilocularis* isolates from different origins [16,17,18]. Bretagne and coworkers (1996) genotyped human AE lesions based on the microsatellites present in the U1snRNA gene spacers [19]. Based on this work, human AE lesions were classified into three profiles according to their geographic origin, with all European patients clustering together in a single profile [16]. The molecular marker EmsB has been extensively used over the last decade to describe the genetic diversity of *E. multilocularis*. EmsB is a tandemly repeated microsatellite with a (CA)n (GA)n pattern [18], located on chromosome 5 of the parasite [20]. This highly polymorphic marker has been characterized [18,21] and used to describe the genetic diversity of the parasite worldwide [21] and in Europe [22], at the level of an individual country [23,24,25,26] and at the local scale [27,28,29,30]. For example, at the European scale, 32 profiles were described (G1 to G32) from 571 adult worms isolated from 123 red foxes [22]. At the scale of a country, 383 adult worms were isolated from 128 red foxes in France, leading to the description of a total of 22 profiles (p1 to p22 in Umhang and coworkers, 2014) [26]. At the local scale, six profiles (α to ζ) were identified in analyses of 140 adult worms isolated from 25 red foxes in Northern France [29]. EmsB was applied on human samples for the marker development [18,21] and for the genotyping of a unique isolate [31]. Through the EWET (EmsB Website for *Echinococcus* Typing) project, a collection of 1211 genotyped European samples, for which information is available concerning geographic location, sampling date and host, has been developed and is continually being improved by the scientific community [32].

The aim of this pioneer study was to use for the first time the molecular marker EmsB to assess the genetic diversity and characterize human *E. multilocularis* infection events temporally and spatially. Robust quality control was applied to the data generated from human alveolar echinococcosis lesions in this study. Moreover, human lesions from different European countries were assigned genetically, with a final comparison with EWET reference data and the generation of graphical maps.

## 2. Results

### 2.1. PCR Conditions

EmsB analysis was improved by modifying PCR conditions relative to previous studies [21,26]. Two mixtures (Multiplex PCR Master Mix (MLX) and Platinum Taq Polymerase (PL)) were compared for four AE lesions from a panel of 120 samples referred for PCR diagnosis of *Echinococcus* infection (Figure 1). The four sets of Multiplex PCR master mix conditions (MLX) results (conditions A to D) obtained for EmsB were compared with those obtained with Platinum Taq DNA polymerase mixture (PL) by Euclidean distance calculation. The smallest distance between MLX and PL EmsB results for the four samples was obtained for MLX condition A and PL (summary in Table 1 and distance matrix in Appendix A).

Following this testing of PCR conditions, we adopted MLX condition A (3% DMSO and a final concentration of 0.1 µM for the primers) for amplification of the AE tested samples, and we further checked EmsB profile stability by performing PL PCR in parallel to MLX PCR. From the 120 AE samples initially tested, 66 samples isolated from 63 patients provided results in the two conditions (Table 2). The panel was composed of 66 surgically resected AE lesions from 63 patients, with 59 frozen samples (44 from liver surgery, 11 from extrahepatic tissues (bone (*n* = 5), brain (*n* = 1), muscles (*n* = 1), diaphragm (*n* = 1), lung (*n* = 1), spleen (*n* = 1), other tissue (*n* = 1)) and non-specified (*n* = 4), stored at −20 °C until analysis) and 7 formalin-fixed and paraffin-embedded tissues (FFPE) (five from liver surgery and two from non-specified tissues). One patient provided two independent samples (frozen and FFPE liver samples) and one patient provided three independent samples (fresh bone samples) isolated during different surgical interventions (Table 2). The patients underwent surgery in four European countries: in six regions and 15 *départements* (a French administrative unit) in France (*n* = 45), seven *cantons* in Switzerland (*n* = 15), two *Ländern* in Germany (*n* = 5), and in Belgium (*n* = 1). The specimens were provided by the following centers: Henri Mondor Hospital (HMH, Assistance Publique - Hôpitaux de Paris (APHP), France), the Institute of Parasitology of Berne (IPA, Vetsuisse Faculty, Switzerland) and the National Reference Centre for Echinococcoses (NRC-E, Besançon University Hospital, France) (Table 2). The mean age of the patients at surgery was 55.8 years (95% CI: 51.6–60.0).

A distance matrix was generated (Appendix A) and a dendrogram was built to check the EmsB results obtained with the two PCR protocols (Appendix A). For the 66 samples, 62 MLX-PL PCR pairs with a genetic distance of less than 0.1 were clustered together in the dendrogram (94% of the panel). The four poorly assembled samples presented PL profiles in which the samples clustered outside the fixed genetic threshold. For seven samples, EmsB profiles were obtained with formalin-fixed paraffin-embedded (FFPE) material. Six of these PCR pairs were correctly clustered under the fixed threshold. The samples 34-HP-122282-GE-Ber (FFPE) and 35-HP-122285-GE-Ber (fresh material) originated from the same patient and clustered together. For FFPE sample 15-HP-061502-GE-Bw, the two PCR clustered separately in the dendrogram, but with an individual distance of 0.082 between the MLX-A and PL conditions (Appendix A).

The EmsB calibrator for fragment size analyses (FSA) run with each EmsB PCR was within the expected size range for all the analyses performed: three peaks at 188, 190 and 192 bp.

### 2.2. Interlaboratory Control Test

The FSA results obtained for the EchinoRisk samples at the IGB (Institute of Genetics of Berne, Switzerland) and the NRC-E laboratories were compared. A 2 bp shift was observed between the two laboratories (Figure 2). A –2 bp correction was therefore applied to the NRC-E results to ensure that all results were within the reference range [32]. Euclidean distances were calculated between the results obtained at the two laboratories, with the correction applied (Table 3). For a given EchinoRisk tested sample, the MLX-A, PL and IGB conditions were applied (Appendix A). The Euclidean distances were below the applied threshold of 0.1 for all but one of the 14 samples (sample UCPR1-499), which gave a stutter-band profile in IGB conditions (Figure 2).

### 2.3. Hierarchical Clustering Analyses and Distribution of EmsB Profiles

Hierarchical clustering analyses were performed on the 66 DNA samples (Table 2) and outgroups. A threshold genetic distance of 0.1 was used for the dendrogram. Sixty-two patients were clustered into nine profile groups, named P1 to P9 and presented in Figure 3, with the remaining four patients not assigned to any of these groups under the genetic distance threshold of 0.1. The P4 profile clustered 29% (19 patients) of the total samples, and the P8 profile clustered 14% (9 patients) of the total samples. The geographic distribution of the nine profiles is shown in Figure 4a for all profiles, whereas Figure 4b shows the profiles present on the French-Swiss border only. Six of the nine profiles clustered patients from neighboring administrative Régions (P2, P3, P5, P6, P7 and P9); four clustered only patients from French *départements* (P2, P3, P5 and P7), one clustered only patients from Swiss cantons (P6) and one clustered patients from neighboring Swiss cantons and French *départements* (P9). Three profiles clustered patients from non-adjoining Régions (P1, P4 and P8). For example, the P1 profile clustered patients from Berlin Bundesland (Germany), the Brussels Région (Belgium), the Canton of Valais (Switzerland) and three French *départements*. The P4 profile clustered patients from six French *départements* (Ain, Pas-de-Calais, Haute-Saône, Jura, Doubs and Territoire de Belfort), Freiburg and Baden-Württemberg (Germany) and the Canton of Berne (Switzerland), whereas the P8 profile clustered nine French patients from nine different French *départements*, with two of the patients located 900 km apart (Figure 4a). The nine patients with extrahepatic lesions presented the profiles P1 (*n* = 1), P4 (*n* = 3), P6 (*n* = 2), P8 (*n* = 2) and one not assigned to any of these groups. The profile P6 clustered three bone tissue isolates from the same patient.

Figure 5 provides graphical documentation of the occurrence of the identified profiles between 1986 and 2018. Each specific profile appeared one to four times, in different patients, in a given year of surgery.

### 2.4. Similarities between Individuals

Individual genetic distances were calculated between the tested samples and the EWET reference data. The most similar reference samples (Euclidean distance of 0 to 0.1) were classified with the tested samples (Appendix A). The geographic distribution of samples and reference specimens was plotted graphically on maps (Figure 6a,b). For example, for the local EmsB profiles (P2, P3, P5, P6, P7 and P9) and widely distributed profiles (P1, P4 and P8), one representative sample was graphically represented for each profile (Figure 6b). Patients presenting local or widely distributed profiles were mapped together with the matching reference samples.

Information about spatial and temporal changes in place of residence or for trips to epidemiologically different areas was available for 35 patients. Four of these patients lived outside areas of high endemicity, and one patient declared not having lived in or traveled to an endemic area.

The four patients with genotypes not corresponding to any of those found in other patients had genotypes similar to reference genotypes, mostly from foxes (Appendix A).

## 3. Discussion

This study included only data on human AE that had passed a prior quality control process. This preselection was a prerequisite for data analysis, given the complexity of the EmsB marker and the nature of the FSA. We added DMSO to the PCR mixture to improve the stability of the EmsB results as suggested by Baskaran et al. (1996) and Jensen et al. (2010) [33,34]. Moreover, the use of an internal calibrator appeared to be essential, given that different sequencing machines were used. For a robust quality control process, we recommend these conditions: (i) the use of two PCR conditions for each sample tested; (ii) calculation of the Euclidean distance between the two FSA obtained from the two PCRs; and (iii) the use of an internal calibrator. For validation of this process, we investigated DNA from samples from the same patient conserved in different ways (frozen and FFPE tissue specimens). No differences were found between the two samples from the same patient. We therefore concluded that FFPE specimens could be used for retrospective genetic studies, even with a multilocus microsatellite. 

We re-evaluated the genetic threshold usually retained for EmsB analysis for the description of profiles from the hierarchical clustering analysis. Based on the hierarchical clustering analysis, we set the threshold at 0.1, which made it possible to define nine relevant profiles on the basis of graphical differences between EmsB electrophoregrams. The threshold generally used for such analyses (0.08) would have distinguished 13 profiles and seven unclassified samples, which would almost certainly have constituted an over-discrimination between samples.

All the EmsB genotype profiles obtained for patients in this study had already been described in the EWET reference collection, and we obtained no entirely new profiles. Groups of patients living in the same geographic area presented identical or similar profiles. As highlighted by the individual research of similarity and mapping (Figure 6b, Appendix A) [22,31], some of these profiles had already been described locally in animals (profiles P5, P6 and P9), as locally clustered among patients. Other profiles seemed to be widespread in foxes in Europe (profiles P2, P3, P7). This suggests two issues in contamination. These patients living in areas of high endemicity may have been contaminated in their residential environment, either with profiles circulating locally or with profiles more widespread within Europe. A similar pattern of profile mixtures and distributions has already been described in Europe [22]. Some profiles were widely distributed throughout Europe, whereas other profiles had more restricted local distributions. The European profiles G05, G21 and G23 [22] predominated numerically in previous studies and were highly widespread. In this study, samples of human origin in the profiles P1-P2-P3, P4-P5 and P7-P8, respectively, could be traced back to these profiles. 

Overall, it was very complex to classify samples accurately to a given profile based on EmsB hierarchical clustering analysis, using only the dendrogram and a genetic threshold. This is due to the nature of the marker itself and to the UPGMA method used to cluster the samples, based on an arithmetic mean and the classification depending on the pre-existing similarity among the samples investigated or added to the analysis. However, despite the limitations of this method, it did make it possible to describe the diversity within a given set of samples. In our previous studies, based on the global shape of the EmsB electrophoregram (number and position of the peaks),“assemblages” were described [29] in which diversity was associated with different profiles. The application of this concept to the results obtained in this study resulted in the description of six assemblages: profiles P1-P2-P3 could be grouped into one assemblage, profiles P7-P8 into another one, and the profiles P4, P5, P6 and P9 represented four different assemblages. It seems interesting to focus on profiles or assemblages, depending on the question and the geographical scale taken into consideration.

Based on analyses of similarity between individuals and mapping, we were able to identify the samples most similar to the tested isolate. By contrast, the dendrogram represents the diversity between the present human samples by highlighting the different profiles existing within the collection. Profiles P4 and P5 were both similar to the European G21 profile. However, the distribution of the P5 profile and associated EWET reference samples seemed to be geographically more limited than the distribution of the numerically dominant profile P4 (Figure 6b), suggesting the possibility of a local profile drifting from a major profile. Based on hierarchical clustering analysis, the lesions of four patients could not be clustered with any other human AE human lesions in this specimen collection. However, searches for similarities between individual samples made it possible to match these samples with EWET reference samples from animals. This result highlights the importance of using two classification approaches (dendrogram based on hierarchical clustering analysis and searches for individual similarity based on sorted lists from distance matrix). More EmsB data should also be collected, particularly given that some of the profiles seem to be rare or did not match other samples in the dendrogram approach. The genetic diversity of EmsB in Europe is undoubtedly greater than currently thought using sequencing of few mitochondrial genes [13,14,15]. This study brings the number of genotyped parasite samples in the EmsB data collection to more than 1300.

Based on this large amount of data, we can now speculate, to some extent, on the conditions in which contamination occurs when hosts encounter parasites. Patients living in endemic areas repeatedly come into contact with locally circulating *E. multilocularis* isolates, but not necessarily with the numerically dominant profiles. The P8 profile, for example, was described in five patients living outside of areas of high endemicity. This profile corresponded to the previously described G23 profile, the second most prevalent profile in Europe [23]. It remains unclear why this profile grouped together most of the patients not living in endemic areas in France. The limits of the area of endemicity in France may need to be reconsidered. Nine patients presenting extrahepatic lesions presented four different profiles. One of these profiles, P4 accounted for three patients. It will be interesting to investigate in greater detail the association between specific profiles and specific organ locations as well as lesion numbers of the parasite in a larger sample of patients. The PNM system (P = parasitic mass in the liver, N = involvement of neighbouring organs, and M = metastasis) permits the clinical classification of alveolar echinococcosis [35,36] and it could be relevant to compare to EmsB profiles. Unfortunately, we only had available in the present study one third of the PNM data for the studied patients. Due to the lack of data we decided to deal with this subject in a future study.

Microsatellite DNA has a high mutation rate which is more difficult to assess for multilocus microsatellites, such as the EmsB marker. However, in this study certain profiles were detected in patients over a period of 30 years, as shown for the P4 profile. Locally, the P6 profile persisted over a period of 10 years and similar profiles were described in foxes. Despite the complex nature of EmsB, we gained insight into the apparent persistence of *E. multilocularis* EmsB profiles in the environment. However, EmsB analyses in the various definitive (i.e., foxes, raccoon dogs and domestic dogs) and intermediate hosts (including aberrant intermediate hosts such as primates, pigs and others) will be required to characterize the fluctuating spatiotemporal presence of *E. multilocularis* in more detail. A large whole-genome sequencing project for *E. multilocularis* specimens has also been proposed. Next-generation sequencing techniques could be used to obtain coding sequence data from the genomic DNA or mitochondrial genome, and non-coding DNA sequence data for microsatellites and transposons, from large collections of samples. Given the relatively low level of polymorphism observed among *E. multilocularis* specimens [15,37,38], studies of all the various types of DNA are likely to be required to elucidate the putative correlation between genetic diversity and potential pathogenicity in humans and animals.

## 4. Conclusions

The genetic diversity of *E. multilocularis* parasite isolates from European human AE patients was assessed for the first time with the highly polymorphic EmsB microsatellite marker. This genetic diversity was compared to the EWET collection of reference, mostly composed of parasite specimens from foxes. Patients living in a highly endemic area presented common EmsB profiles. These profiles were described in foxes in a limited geographical area for some of them or largely in Europe for others. Moreover, some EmsB profiles were described among patients over a period of 30 years. Thanks to this study, patients and animals were described as basically sharing the same EmsB profiles in Europe. Even if considered as an aberrant and dead-end host, this present work allowed us to document the indirect involvement or position of humans within the *E. multilocularis* parasite life cycle. With regards to these findings, one can speculate no intermediate host selection is achieved by the parasite strains.

## 5. Materials and Methods 

### 5.1. Alveolar Echinococcosis Tissue Collection and DNA Extraction

An initial panel of 120 AE samples from surgery were collected between 1981 and 2019 and referred for PCR diagnosis of *Echinococcus* infection. The specimens were provided by the Henri Mondor Hospital (HMH, Assistance Publique - Hôpitaux de Paris (APHP), France), the Institute of Parasitology of Berne (IPA, Vetsuisse Faculty, Switzerland) and the National Reference Centre for Echinococcoses (NRC-E, Besançon University Hospital, France). A control outgroup was constituted from three more *E. multilocularis* samples—one from Hokkaido, Japan, (*n*=1), provided by the Asahikawa Medical University (Hokkaido, Japan), one from Alaska, United States (*n*=1), provided by the Institute of Parasitology of Berne, and one from China (*n*=1), provided by the Chrono-environment Laboratory (University of Bourgogne Franche-Comté (UBFC), France)—together with one *E. granulosus* sensu stricto sample (originating from an Algerian sheep) from the Chrono-environment Laboratory [21]. For the NCR-E collection, information about patients, such as place of residence and endemicity at this site, as well as change in residence and/or trips to other endemic areas, were obtained from the FrancEchino database (NRC-E). 

Biological material was obtained for standard diagnosis on the basis of the physicians’ prescriptions. Data were rendered anonymous for analysis. For the ethic statement in this study, according to French Public Health Law [39], protocols of this type do not require approval from an ethics committee and are exempt from the requirement for formal informed consent.

At the IPA, DNA from AE lesions was purified with the QIAamp DNA Mini kit (Qiagen, Hilden, Germany). At the NRC-E, DNA was purified with the High Pure PCR Template Preparation kit (Roche Diagnostics, Mannheim, Germany). For FFPE, DNA was purified with the QIAamp DNA FFPE tissue kit (Qiagen, Hilden, Germany), in accordance with the manufacturer’s instructions, at both institutions. At Henri Mondor Hospital, DNA was extracted from a collection of cysts, by cetyltrimethylammonium bromide (CTAB) precipitation [16], as previously described [40].

### 5.2. PCR Conditions

For the Henri Mondor Hospital collection, PCR analyses were performed at the Institute of Genetics, University of Berne (IGB, Switzerland). For the IPA and NRC-E collections, PCR analyses were performed in the Chrono-environment Laboratory (France). 

In the IGB laboratory, a single set of PCR conditions was applied to the DNA samples. The PCR mixture (final volume: 15 µL) contained 200 µM of each deoxynucleoside triphosphate (GeneAmp dNTPs; Applied Biosystems, Foster City, CA, USA), 0.4 µM forward primer 5’-labeled with a specific fluorescent dye (EmsB A: 5‘-Fam-GTGTGGATGAGTGTGCCATC-3’), 0.7 µM classical reverse primer (EmsB C: 5’-CCACCTTCCCTACTG-CAATC-3’), and 0.5 U AmpliTaq DNA polymerase enzyme in GeneAmp 1X PCR buffer (10 mM Tris-HCl, pH 8.3, 50 mM KCl, 1.5 mM MgCl_2_, and 0.001% gelatin; Applied Biosystems, Foster City, CA, USA) and up to 50 ng of purified DNA (IGB conditions). 

At the NRC-E, 5 different sets of PCR conditions were applied to four DNA samples (01-HP-102039-BE-Bru, 15-HP-061502-GE-Bw, 42-HP-122345-SW-Fr and BON-363-FR-39) as part of a quality control process for EmsB profile validation. Dimethylsulfoxide (DMSO) and various concentrations of primers were used to assess the stability of EmsB electrophoregrams (Table 4). The first PCR mixture had a final volume of 18 µl and contained the Multiplex PCR master mix with 2.8 U HotStarTaq DNA polymerase (Qiagen, Hilden, Germany), 0.1 or 0.5 µM of each primer (final concentrations; EmsB A and EmsB C primers), 0% or 3% DMSO, and up to 50 ng of purified DNA (MLX conditions A to D). The second PCR mixture contained (in a final volume of 25 µL): 2 U Platinum Taq DNA polymerase (Invitrogen, Carlsbad, CA, USA), 1X PCR buffer, 0.2 mM of each dNTP (Invitrogen, Carlsbad, CA, USA), 1.5 mM MgCl_2_, as recommended by the manufacturer, 0.2 µM of each primer (final concentration; EmsB A and EmsB C primers), 3% DMSO and up to 50 ng of purified DNA (PL conditions). Once the optimal MLX conditions had been defined, each sample from the IPA and NRC-E collections was tested in both the chosen MLX and PL conditions. For each run, the EmsB calibrator—a plasmid construct containing four EmsB microsatellites—was used to check and compare FSA reliability [32]. The PCR conditions used are summarized in Table 5.

All PCRs were performed with a Biometra T3 thermocycler (Whatman Biometra, Göttingen, Germany). 

FSA was performed with fluorescently labeled PCR products by capillary electrophoresis on an ABI PRISM 3100 Genetic Analyzer (Applied Biosystems, Foster City, CA, USA) for the Henri Mondor collection and on an Applied Biosystems 3130 Genetic Analyzer for the IPA and NRC-E collections. Electrophoregrams were analyzed with GeneMapper 4.1 (Life Technologies, CA, USA).

### 5.3. Hierarchical Clustering Analysis

With the FSA technique, the electrophoregrams for the EmsB target are presented as a series of peaks corresponding to alleles [18,20]. The presence or absence of peaks and the heights of the associated fluorescence intensity peaks were recorded as previously described [32]. These data were used to assess the genetic diversity of the parasite and to establish profiles by hierarchical clustering analysis using Euclidean distance and the unweighted pair group method with arithmetic mean (UPGMA) [41]. A multiscale bootstrap resampling (B = 1000) was performed to assess the stability of the clusters, resulting in approximately unbiased *p*-values [42,43]. Clustering analyses were performed with RStudio software (R version 3.5.1) [44] and the pvclust package [45]. The genetic distance threshold previously reported by Knapp and co-workers [21] and applied to the dendrogram to describe clusters or profiles in collections of samples was challenged here. The original threshold was obtained by calculating the mean (x) genetic distance between three samples from a single strain maintained in vivo by three successive transperitoneal inoculations in *Meriones unguiculatus*, plus 3 standard deviations (σ) according to the formula x + 3 σ, giving a genetic distance threshold of 0.08. This threshold calculation was modified (x + 4 σ), yielding a genetic distance threshold of 0.1. This modification was applied to the assessment of genetic diversity in *E. multilocularis* to reduce the likelihood of over-discrimination by this method on samples stored in different conditions, obtained on different dates and processed with different machines. MLX and PL conditions were compared by using the resulting FSA to generate a first dendrogram. The Euclidean distance between the two conditions was checked to assess the stability of the FSA obtained from EmsB PCR products. With the best conditions, a dendrogram with *E. multilocularis* human cases only was generated for a better graphic rendering.

### 5.4. EWET Collection of Reference and Individual Research of Similarity

The EmsB genotyping results obtained for these AE patient lesions were compared to data referenced in database of the EmsB Website for the *Echinococcus* typing project (EWET project) [32], for 1211 genotyped EmsB samples from 13 European countries, isolated from adult worms (fox, cat, and raccoon dog hosts) and metacestodes (human, monkey, and rodent hosts) (Figure 6a and Table 6) [18,21,22,23,26,27,29,46,47]. Patients and EWET reference data were mapped with Quantum GIS software version 3.6.0 (QGIS, Open Source Geospatial Foundation Project. http://qgis.osgeo.org), and Eurostat map bases (https://ec.europa.eu/eurostat/fr/web/gisco/geodata/reference-data/administrative-units-statistical-units/countries). Our samples were compared with the data collection to check for individual similarities. The most similar EWET reference samples can be obtained with R software as a sorted list, following EmsB Guidelines section VII [32]. The geographic distribution of the EWET reference samples associated with a tested patient sample was represented with QGIS.

### 5.5. Interlaboratory Control Test

The EmsB data were generated by two laboratories. We therefore checked the reproducibility of the technique between the two laboratories. Fourteen *E. multilocularis* worms from three foxes studied in the EchinoRisk project [22] were analyzed at the two laboratories (PAHO5-117 to 120 from a German fox, UCPR1-495 to 499 from a Czech fox, and 16PL-375 to 379 from a Polish fox) under the PCR conditions described above (AmpliTaq PCR mixture in the IGB and MLX and PL best conditions in the NRC-E), and the FSA data were compared by Euclidean distance calculation.

## Figures and Tables

**Figure 1 pathogens-09-00282-f001:**
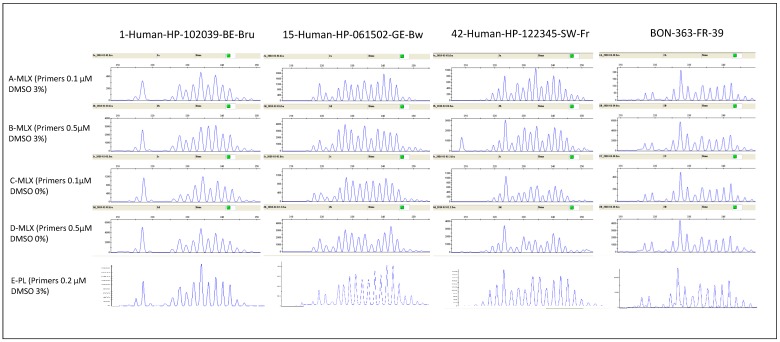
Electrophoregrams for genotyping performed on four human alveolar echinococcosis (AE) samples for the four Multiplex mixture conditions (MLX-A to D) and the Platinum mixture (PL).

**Figure 2 pathogens-09-00282-f002:**
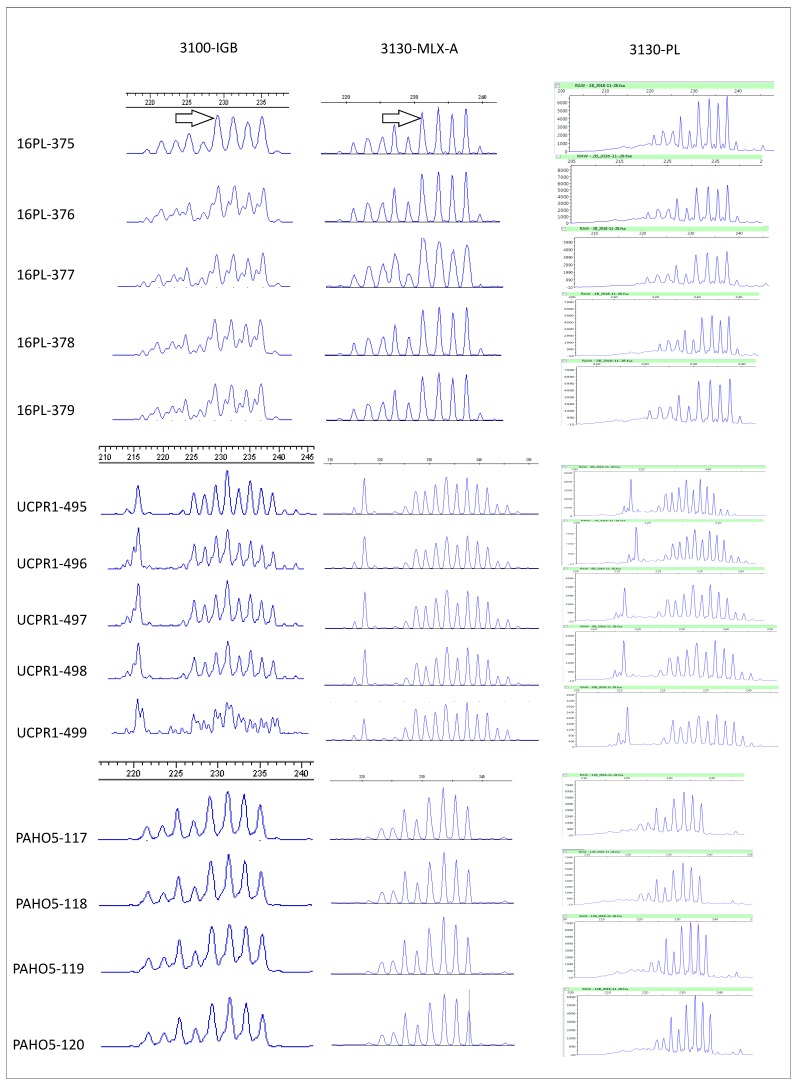
Electrophoregrams for the genotyping performed on 14 EchinoRisk samples from three foxes in Multiplex PCR master mix (Qiagen) conditions (MLX A to D) and Platinum Taq DNA polymerase mixture (Invitrogen) conditions (PL) at the Besançon Laboratory (NRC-E) and in AmpliTaq DNA polymerase mixture (Applied Biosystems) conditions at the Berne Laboratory (IPA). The arrows indicate the 2 bp shift observed between the results obtained with the ABI PRISM 3100 Genetic Analyzer and the Applied Biosystems 3130 Genetic Analyzer.

**Figure 3 pathogens-09-00282-f003:**
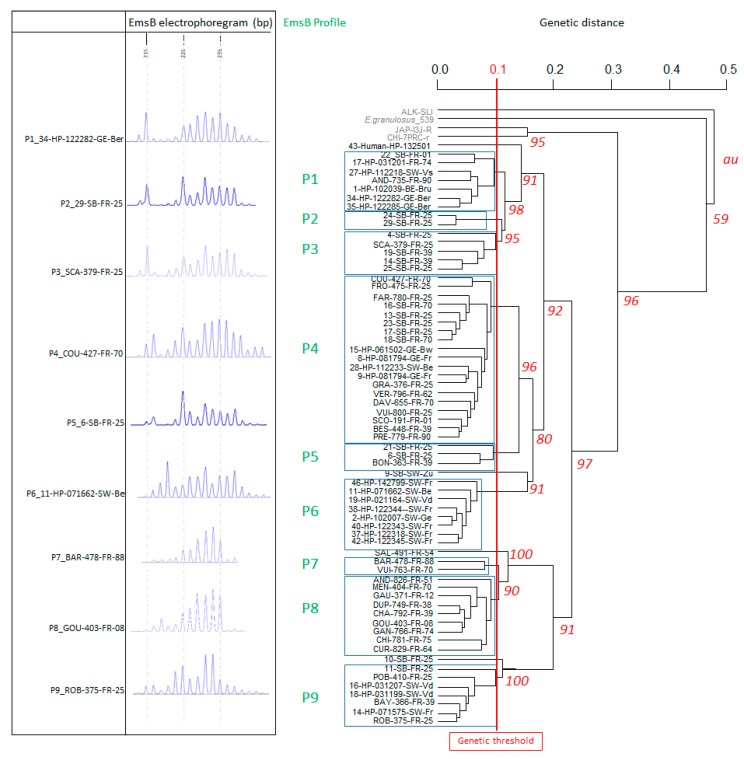
Dendrogram constructed from EmsB amplification data by hierarchical clustering analysis (Euclidean distance and unweighted pair group method). Approximately unbiased (*au*) *p*-values are indicated at tree nodes as percentages calculated by multiscale bootstrapping (B = 1000). *E. granulosus* sensu stricto (G1 strain, originating from an Algerian sheep), and three *E. multilocularis* from Alaska (ALK-SLI), Japan (JAP-I3J-R) and China (CHI-7PRC-r) were used as control outgroups. A genetic distance threshold of 0.1 was used to distinguish EmsB profiles. A representative EmsB electrophoregram is provided for each profile.

**Figure 4 pathogens-09-00282-f004:**
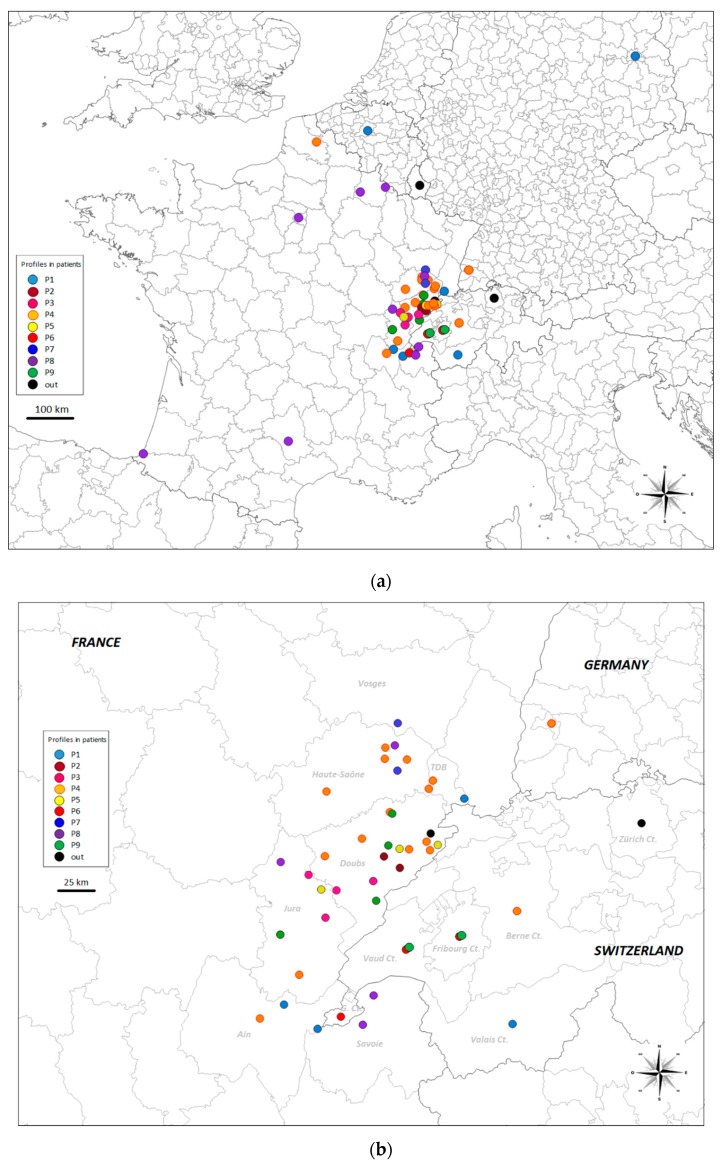
Distribution of the EmsB profiles described for human AE lesions in the countries studied (**a**), and French endemic Région profiles (**b**). TDB: Territoire de Belfort; Ct.: Canton; G. Ct.: Canton of Geneva.

**Figure 5 pathogens-09-00282-f005:**
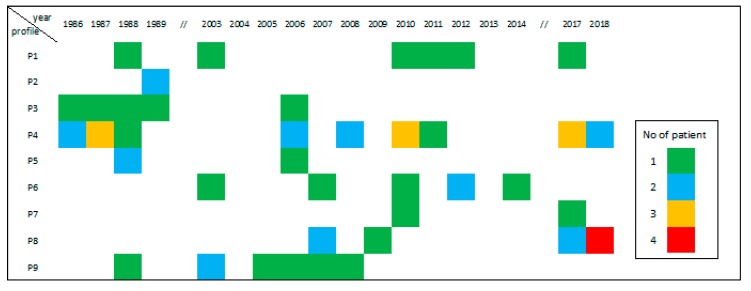
Distribution over time, from 1986 to 2018, of the nine EmsB profiles (P1 to P9) for the 63 alveolar echinococcosis patients genotyped.

**Figure 6 pathogens-09-00282-f006:**
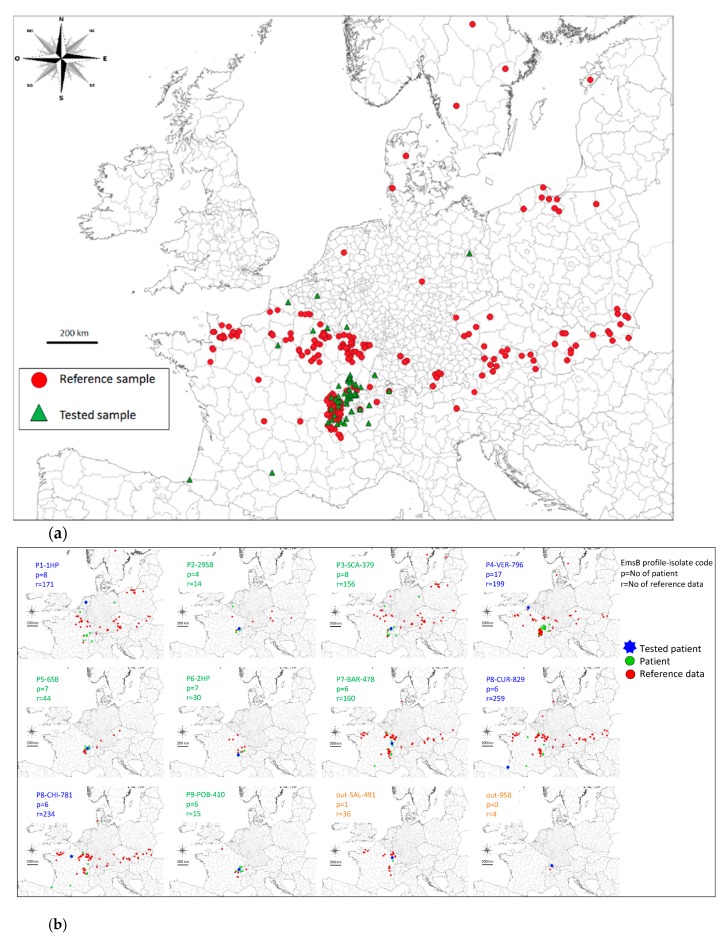
Distribution of the 66 AE human samples and the 1211 EWET (EmsB Website for *Echinococcus* Typing) reference data (**a**), and graphical representation of the EmsB profiles described for human AE lesions and similar EWET reference data selected on the basis of a Euclidean distance of less than 0.1 to the tested samples (**b**).

**Table 1 pathogens-09-00282-t001:** Euclidean distances between genotyping results for four human AE samples obtained by PCR in Multiplex mixture conditions (MLX A to D) and Platinum mixture conditions (PL).

MLX/PL	1HP-PL	BON-PL	15HP-PL	42HP-PL
1HP-MLX-A	0.0518			
1HP-MLX-B	0.0684			
1HP-MLX-C	0.0706			
1HP-MLX-D	0.0769			
BON-MLX-A		0.0488		
BON-MLX-B		0.0548		
BON-MLX-C		0.0524		
BON-MLX-D		0.0637		
15HP-MLX-A			0.0829	
15HP-MLX-B			0.1036	
15HP-MLX-C			0.1024	
15HP-MLX-D			0.1050	
42HP-MLX-A				0.0618
42HP-MLX-B				0.0737
42HP-MLX-C				0.0752
42HP-MLX-D				0.0966
SD	0.0107	0.0063	0.0104	0.0145

Samples 1HP: 1-HP-102039-BE-Bru; BON: BON-363-FR-39; 15HP: 15-HP-061502-GE-Bw; 42HP: 42-HP-122345-SW-Fr.

**Table 2 pathogens-09-00282-t002:** Human alveolar echinococcosis panel and main characteristics.

Code	Panel	Country	Place of Residence	Organ of Origin	Type of Biological Material	Year of Surgery	Age at Surgery	Sex	EmsB Profile	Living in Endemicarea	Moved to Endemic Région	Trip to Endemic Région
01-HP-102039-BE-Bru	IPA	Belgium	Brussels Hoofdstedelijk Gewest	ND	FFPE	2010	75	M	P1	Yes	ND	ND
04-SB-123-FR-25 ^3^	HMH	France	BFC Région, Doubs Dép., Amancey Ct.	Liver tissue	Fresh	1986	61	F	P3	Yes	No move	No trip
06-SB-289-FR-25 ^3^	HMH	France	BFC Région, Doubs Dép., Pierrefontaine-les-V. Ct.	Liver tissue	Fresh	1988	58	M	P5	Yes	ND	ND
10-SB-125-FR-25 ^3^	HMH	France	BFC Région, Doubs Dép., Saint-Hippolyte Ct.	Liver tissue	Fresh	1986	65	M	out	Yes	No move	No trip
11-SB-209-FR-25 ^3^	HMH	France	BFC Région, Doubs Dép., Pierrefontaine-les-V. Ct.	Liver tissue	Fresh	1988	55	F	P9	Yes	No move	ND
13-SB-HFE-FR-25 ^3^	HMH	France	BFC Région, Haute-Saône Dép., Besançon Ct.	Liver tissue	Fresh	1988	15	F	P4	Yes	ND	ND
14-SB-164-FR-39 ^3^	HMH	France	BFC Région, Jura Dép., Lons-le-Saunier Ct.	Liver tissue	Fresh	1987	38	M	P3	Yes	Yes	No
16-SB-154-FR-70 ^3^	HMH	France	BFC Région, Haute-Saône Dép., Luxeuil-les-Bains Ct.	Liver tissue	Fresh	1987	63	F	P4	Yes	ND	ND
17-HP-354-FR-74	IPA	France	ARA Région, Hte-Savoie Dép., Saint-Julien-en-G. Ct.	Liver tissue	Fresh	2003	61	F	P1	Yes	Yes	Yes
17-SB-487-FR-25 ^3^	HMH	France	BFC Région, Doubs Dép., Roulans Ct.	Liver tissue	Fresh	1987	63	M	P4	Yes	No move	ND
18-SB-230-FR-70 ^3^	HMH	France	BFC Région, Haute-Saône Dép., Héricourt-Est Ct.	Liver tissue	Fresh	1986	24	F	P4	Yes	No move	yes
19-SB-244-FR-39 ^3^	HMH	France	BFC Région, Jura Dép., Champagnole Ct.	ND	Fresh	1988	54	M	P3	Yes	No move	No trip
21-SB-253-FR-25 ^3^	HMH	France	BFC Région, Doubs Dép., Maîche Ct.	ND	Fresh	1988	44	M	P5	Yes	No move	No trip
22_SB-239-FR-01 ^3^	HMH	France	ARA Région, Ain Dép., Oyonnax-Sud Ct.	Liver tissue	Fresh	1988	52	M	P1	Yes	No move	No
23-SB-274-FR-70 ^3^	HMH	France	BFC Région, Haute-Saône Dép., Fresne-Saint-M. Ct.	Liver tissue	Fresh	1986	36	F	P4	Yes	No move	No trip
24-SB-251-FR-25 ^3^	HMH	France	BFC Région, Doubs Dép., Pierrefontaine-les-V. Ct.	Liver tissue	Fresh	1989	53	F	P2	Yes	No move	No trip
25-SB-94-FR-25 ^3^	HMH	France	BFC Région, Doubs Dép., Quingey Ct.	ND	Fresh	1989	51	M	P3	Yes	ND	ND
29-SB-FR-25 ^3^	HMH	France	BFC Région, Doubs Dép.	Liver tissue	Fresh	1989	58	F	P2	Yes	ND	ND
AND-735-FR-90	UHCB	France	BFC Région, Territoire-de-Belfort Dép., Delle Ct.	Extrahepat. Mass	Fresh	2017	71	M	P1	Yes	No move	No trip
AND-826-FR-51	UHCB	France	Grand-Est Région, Ardennes Dép., Château-Porcien	Liver tissue	Fresh	2018	36	M	P8	Yes	Yes	Yes
BAR-478-FR-88	UHCB	France	Grand-Est Région, Vosges Dép., Le Val-d’Ajol Ct.	Liver tissue	Fresh	2010	36	F	P7	Yes	Yes	Yes
BAY-366-FR-39	UHCB	France	BFC Région, Jura Dép., Lons-le-Saunier Ct.	Liver tissue	Fresh	2005	67	M	P9	Yes	No move	Yes
BES-448-FR-39	UHCB	France	BFC Région, Jura Dép., Moirans-en-Montagne Ct.	Liver tissue	Fresh	2010	68	M	P4	Yes	No move	Yes
BON-363-FR-39	UHCB	France	BFC Région, Jura Dép., Arbois Ct.	Liver tissue	Fresh	2006	22	M	P5	Yes	No move	Yes
CHA-792-FR-39	UHCB	France	BFC Région, Jura Dép., Authume Ct.	Liver tissue	Fresh	2018	63	M	P8	Yes	Yes	Yes
CHI-781-FR-75	UHCB	France	Ile-de-France Région, Paris Dép.	Bone tissue	Fresh	2017	67	F	P8	No	Yes	No
COU-427-FR-70	UHCB	France	BFC Région, Haute-Saône Dép., Luxeuil-les-Bains Ct.	Liver tissue	Fresh	2010	41	M	P4	Yes	No move	Yes
CUR-829-FR-64	UHCB	France	NA Région, Pyrénées-Atlantiques Dép., Bayonne Ct.	Diaphragm	Fresh	2018	69	M	P8	No	No	Yes
DAV-655-FR-70	UHCB	France	BFC Région, Haute-Saône Dép., Mélisey Ct.	Liver tissue	Fresh	2017	65	F	P4	Yes	ND	ND
DUP-749-FR-74	UHCB	France	ARA Région, Haute-Savoie Dép., Sciez Ct.	Liver tissue	Fresh	2017	70	M	P8	Yes	No move	Yes
FAR-780-FR-25	UHCB	France	BFC Région, Doubs Dép., Maîche Ct.	Lung tissue	Fresh	2017	72	F	P4	Yes	No move	No trip
FRO-475-FR-25	UHCB	France	BFC Région, Doubs Dép., Baumes-les-Dames Ct.	Liver tissue	Fresh	2010	53	F	P4	Yes	No move	No trip
GAN-766-FR-74	UHCB	France	ARA Région, Haute-Savoie Dép., La Roche-sur-F. Ct.	Liver tissue	Fresh	2018	28	F	P8	Yes	ND	ND
GAU-371-FR-12	UHCB	France	Occitanie Régions, Aveyron Dép., Raspes-et-L. Ct.	Liver tissue	Fresh	2007	31	M	P8	No	No move	No
GOU-403-FR-08	UHCB	France	Grand-Est Région, Ardennes Dép., Carignan Ct.	Liver tissue	Fresh	2009	24	M	P8	Yes	No move	No trip
GRA-376-FR-25	UHCB	France	BFC Région, Doubs Dép., Valdahon Ct.	Liver tissue	Fresh	2006	73	M	P4	Yes	Yes	Yes
MEN-404-FR-70	UHCB	France	BFC Région, Haute-Saône Dép., Mélisey Ct.	Liver tissue	Fresh	2007	72	F	P8	Yes	No move	No trip
POB-410-FR-25	UHCB	France	BFC Région, Doubs Dép., Pontarlier Ct.	Liver tissue	Fresh	2008	73	M	P9	Yes	Yes	Yes
PRE-779-FR-90	UHCB	France	BFC Région, Territoire-de-Belfort Dép., Bavilliers Ct.	Liver tissue	Fresh	2017	80	M	P4	Yes	ND	ND
ROB-375-FR-25	UHCB	France	BFC Région, Doubs Dép., Bavans Ct.	Liver tissue	Fresh	2006	59	F	P9	Yes	No move	No trip
SAL-491-FR-54	UHCB	France	Grand-Est Région, Meurthe-et-M. Dép., Haroué Ct.	Bone tissue	Fresh	2019	74	M	out	Yes	Yes	No trip
SCA-379-FR-25	UHCB	France	BFC Région, Doubs Dép., Ornans Ct.	Liver tissue	Fresh	2006	68	M	P3	Yes	No move	Yes
SCO-191-FR-01	UHCB	France	ARA Région, Ain Dép., Saint-Etienne-du-Bois Ct.	Liver tissue	Fresh	1987	58	F	P4	Yes	ND	ND
VER-796-FR-62	UHCB	France	HDF Région, Pas-de-Calais Dép., Béthune Ct.	Liver tissue	FFPE	2018	64	F	P4	No	No move	No trip
VUI-763-FR-70	UHCB	France	BFC Région, Haute-Saône Dép., Lure Ct.	Liver tissue	Fresh	2017	68	F	P7	Yes	ND	ND
VUI-800-FR-25	UHCB	France	BFC Région, Doubs Dép., Maîche Ct.	Splenic lesion	Fresh	2018	69	F	P4	Yes	No move	Yes
08-HP-081794-GE-Bw	IPA	Germany	Baden-Württemberg Land, Freiburg District	Liver tissue	FFPE	2008	55	F	P4	Yes	ND	ND
09-HP-081788-GE-Bw	IPA	Germany	Baden-Württemberg Land, Freiburg District	Liver tissue	FFPE	2008	64	M	P4	Yes	ND	ND
15-HP-061502-GE-Bw	IPA	Germany	Baden-Württemberg Land, Freiburg District	Liver tissue	FFPE	2006	76	F	P4	Yes	ND	ND
34-HP-122282-GE-Ber ^1^	IPA	Germany	Berlin Land	Liver tissue	FFPE	2012	33	M	P1	Yes	ND	ND
35-HP-122285-GE-Ber ^1^	IPA	Germany	Berlin Land	Liver tissue	Fresh	2012			P1	Yes	ND	ND
02-HP-102007-SW-Ge	IPA	Switzerland	Geneva Ct.	Liver tissue	Fresh	2010	23	M	P6	Yes	ND	ND
09-SB-SW-Zu	HMH	Switzerland	Zürich Ct.	ND	Fresh	1981	ND	ND	out	Yes	ND	ND
11-HP-071662-SW-Be	IPA	Switzerland	Berne Ct.	Liver tissue	Fresh	2007	78	F	P6	Yes	ND	ND
14-HP-071575-SW-Fr	IPA	Switzerland	Fribourg Ct.	Liver tissue	Fresh	2007	77	F	P9	Yes	ND	ND
16-HP-031207-SW-Vd	IPA	Switzerland	Vaud Ct.	Liver tissue	Fresh	2003	34	M	P9	Yes	ND	ND
18-HP-031199-SW-Vd	IPA	Switzerland	Vaud Ct.	Liver tissue	Fresh	2003	61	F	P9	Yes	ND	ND
19-HP-021164-SW-Vd	IPA	Switzerland	Vaud Ct.	Liver tissue	Fresh	2003	43	F	P6	Yes	ND	ND
27-HP-112218-SW-Vs	IPA	Switzerland	Valais Ct.	Liver tissue	Fresh	2011	69	M	P1	Yes	ND	ND
28-HP-112233-SW-Be	IPA	Switzerland	Berne Ct.	Brain tissue	Fresh	2011	41	M	P4	Yes	ND	ND
37-HP-122318-SW-Fr	IPA	Switzerland	Fribourg Ct.	Muscle tissue	Fresh	2012	59	F	P6	Yes	ND	ND
38-HP-122344-SW-Fr ^2^	IPA	Switzerland	Fribourg Ct.	Bone tissue	Fresh	2012	64	F	P6	Yes	ND	ND
40-HP-122343-SW-Fr ^2^	IPA	Switzerland	Fribourg Ct.	Bone tissue	Fresh	2012			P6	Yes	ND	ND
42-HP-122345-SW-Fr ^2^	IPA	Switzerland	Fribourg Ct.	Bone tissue	Fresh	2012			P6	Yes	ND	ND
43-HP-132501-SW-Fr	IPA	Switzerland	Fribourg Ct.	Liver tissue	Fresh	2013	55	M	out	Yes	ND	ND
46-HP-142799-SW-Fr	IPA	Switzerland	Fribourg Ct.	ND	FFPE	2014	59	F	P6	Yes	ND	ND

^1,2^ sample from the same patient; ^3^ patients genotyped with the U1snRNA marker by Bretagne et co-workers in 1996; IPA: Institute of Parasitology, Berne, Switzerland; HMH: Henri Mondor Hospital, Paris, France; UHCB: University Hospital Centre, Besançon, France; ARA: Auvergne-Rhône-Alpes; BFC: Bourgogne Franche-Comté; HDF: Hauts-de-France; NA: Nouvelle-Aquitaine; Dép.: Département; Ct.: Canton; FFPE: formalin-fixed paraffin-embedded; ND: no data.

**Table 3 pathogens-09-00282-t003:** Euclidean distances for genotyping results for 14 EchinoRisk samples obtained from three foxes, in Multiplex PCr master mix (Qiagen) conditions (MLX A) and Platinum Taq polymerase mixture (Invitrogen) conditions (PL) at the Chrono-environment Laboratory (NRC-E), and in AmpliTaq DNA polymerase mixture (Applied Biosystems) conditions at the Institute of Genetics of Berne (IGB).

IGB vs. NRC-E	16PL_MLX	16PL_PL	UCPR1_MLX	UCPR1_PL	PAHO5_MLX	PAHO5_PL
16PL_375	0.0563	0.0473				
16PL_376	0.0589	0.0477				
16PL_377	0.0540	0.0415				
16PL_378	0.0512	0.0429				
16PL_379	0.0540	0.0471				
UCPR1_495			0.0657	0.0669		
UCPR1_496			0.0653	0.0643		
UCPR1_497			0.0575	0.0634		
UCPR1_498			0.0582	0.0632		
UCPR1_499			0.1395	0.0838		
PAHO5_117					0.0285	0.0344
PAHO5_118					0.0214	0.0277
PAHO5_119					0.0413	0.0344
PAHO5_120					0.0264	0.0306
SD	0.0028	0.0028	0.0350	0.0087	0.0084	0.0032

**Table 4 pathogens-09-00282-t004:** PCR mixture and conditions for quality control.

Polymerase	DMSO	Primers	Conditions
HotStarTaq (MLX)	3%	0.1 µM	A
		0.5 µM	B
	0%	0.1 µM	C
		0.5 µM	D
Platinum (PL)	3%	0.2 µM	E

**Table 5 pathogens-09-00282-t005:** PCR conditions for the three protocols applied.

	EmsB-AmpliTaq		EmsB-Multiplex				EmsB-Platinum			
PCR Step	Sample		Sample		Calibrator		Sample		Calibrator	
1-pre-amplification	94 °C	13 min	95 °C	15 min	95 °C	15 min	94 °C	2 min	94 °C	2 min
2-denaturation	94 °C	30 s	94 °C	30 s	94 °C	30 s	94 °C	30 s	94 °C	30 s
3-hybridization	60 °C	30 s	60 °C	90 s	60 °C	90 s	60 °C	30 s	60 °C	30 s
4-elongation	72 °C	60 s	72 °C	60 s	72 °C	30 s	72 °C	60 s	72 °C	60 s
5-final elongation	60 °C	45 min	60 °C	30 min	60 °C	30 min	60 °C	45 min	60 °C	45 min
Steps 2 to 4	X45		X40		X25		X45		X25	

**Table 6 pathogens-09-00282-t006:** Country, host and number of EWET reference samples used for genetic comparison.

*Echinococcus Multilocularis* Host							
Country	Fox	Human	Rodent	Cat	Monkey	Raccoon dog	Total
France	554	1	0	1	0	0	556
Switzerland	84	8	11	0	5	0	108
Austria	98	1	0	0	0	0	99
Poland	95	0	0	0	0	0	95
Germany	88	1	0	0	0	0	89
Czech Republic	67	0	0	0	0	0	67
Slovakia	63	0	0	0	0	0	63
Denmark	38	0	0	0	0	0	38
Sweden	34	0	0	0	0	0	34
Norway	0	0	27	0	0	0	27
Italy	17	0	0	0	0	0	17
Estonia	0	0	0	0	0	11	11
*In vivo* culture	0	0	5	0	0	0	5
Netherlands	2	0	0	0	0	0	2
Total	1140	11	43	1	5	11	1211

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
