# Peer review of "Genotyping Echinococcus multilocularis in Human Alveolar Echinococcosis Patients: An EmsB Microsatellite Analysis"

_pathogens, 2020, doi:10.3390/pathogens9040282_

Round 1

Reviewer 1 Report

The MS by Knapp et colleagues focus on genotyping of the larval state of E. multilocularis in humans. Genotyping of more than 1200 samples already has been done/published on field samples, however only very few studies have analyzed probes from infected humans. If this is the first analysis on human probes using the EmsB marker this is pioneer work and warrants high attention. A  great advantage is that a large published data set is available for comparison. Several testing of methods and lab condition have been done.

Points of criticism raised:

  1. Please make clear that this is the first study on human material regarding this marker. If not cite the other work and outline why this study is particular.
  2. In the abstract, please specify that the 1,211 probes are already published and are available on a public domain.
  3. EM has at least three states: Worm, egg, metacestode. Regarding the EmsB data: has it been checked whether the marker is constant in these three states. Has a comparison been done regarding metacestode versus mouse with metacestode versus human data excluding data from worm and egg?
  4. I still have difficulties to understand whether the human cases have been compared to the large data set? Has the large data set been stratified/separated regarding only worm or animal metacestode data? What happens in the hierarchal clustering when the data sets are reduced to these groups (human animal metacestode/worm) ?
  5. What happens to the clustering if the 32 published profile (G1-G32) are included in the analysis? How stable is clustering and how many cases belong to each group?
  6. Is it possible to perform partitioning cluster analyse/ classification/learning experiments to strengthen the data regarding independent categories of the profiles with a linear support vector machine?
  7. Is there data available regarding the PNM, kind of lesion in ultrasound of the lesions?
  8. Is the presence of the marker analyzed completely excluded in humans?
  9. The lesion in humans is more or less completely necrotic. Have there been done experiments on necrotic metacestodic material to exclude artifacts due to high fragmentation of DNA in the necrosis? Is it possible to perform 4bp- Repeats rather than 2bp-tepeats in order to have a higher accuracy?
  10. Please show examples of all the 9 profiles for comparison.

Author Response

RESPONSE TO THE REVIEWER 1

Comments and Suggestions for Authors

The MS by Knapp et colleagues focus on genotyping of the larval state of E. multilocularis in humans. Genotyping of more than 1200 samples already has been done/published on field samples, however only very few studies have analyzed probes from infected humans. If this is the first analysis on human probes using the EmsB marker this is pioneer work and warrants high attention. A great advantage is that a large published data set is available for comparison. Several testing of methods and lab condition have been done.

Points of criticism raised:

  1. Please make clear that this is the first study on human material regarding this marker. If not cite the other work and outline why this study is particular.

In previous studies, human samples were employed for the EmsB development (Bart et al., 2006, Knapp et al., 2007) or genotyping on a unique isolate (Debourgogne et al., 2014). This was added to the manuscript, lines 99-100 “EmsB was applied on human samples for the marker development [18,21] and for the genotyping of a unique isolate [31]”. However, our work represents the first study dedicated to the comparative assessment of the genetic diversity among human E. multilocularis lesions. As suggested by the referee, we add this information: Lines 104-106 « The aim of this pioneer study was to use for the first time the molecular marker EmsB to comparatively assess the genetic diversity and characterize human E. multilocularis infection events temporally and spatially».

  1. In the abstract, please specify that the 1,211 probes are already published and are available on a public domain.

As suggested by the referee, we add this information on the manuscript: Lines 33-35 « This genetic marker was previously tested on a collection of 1,211 European field samples predominantly of animal origin, referenced on a publicly available database ».

  1. EM has at least three states: Worm, egg, metacestode. Regarding the EmsB data: has it been checked whether the marker is constant in these three states. Has a comparison been done regarding metacestode versus mouse with metacestode versus human data excluding data from worm and egg?

A similar EmsB profile can be both described from parasites isolated from definitive (fox) and intermediate (human or rodent) hosts as we can see in this present study and previous works. For the question of the metacestode stage between hosts, profiles described in Swiss rodents were described in some Swiss patients, as this is the case for example for the patient 37-HP-122318-SW-Fr with the profile P6, who shared the same profile with rodents from the same area.

  1. I still have difficulties to understand whether the human cases have been compared to the large data set? Has the large data set been stratified/separated regarding only worm or animal metacestode data? What happens in the hierarchal clustering when the data sets are reduced to these groups (human animal metacestode/worm) ?

In order to have a readable dendrogram, the hierarchical clustering analysis was built with the human patient cases only. When the dendrogram contains over 150 samples it is quite difficult to have a proper reading of the figure. Additional details were given to clarify this question. Lines 443-444: « With the best conditions, a dendrogram with E. multilocularis human cases only was generated, for a better graphic rendering ». The samples were then compared to the whole EWET collection by individual comparison, presented in the Supplementary table S4. This individual comparison allowed us to elaborate the figure 6.

  1. What happens to the clustering if the 32 published profile (G1-G32) are included in the analysis? How stable is clustering and how many cases belong to each group?

Due to the method used to perform the hierarchical clustering analysis, the UPGMA method based on an arithmetic mean, the constituted groups can slightly differ according to the similarity existing between samples added to the analysis. We noticed that previously described profiles can be split in other profiles according to the addition of samples to the analysis. We add information on this point in the manuscript, lines 288-291: « This difficulty is due to the nature of the marker itself and to the UPGMA method used to cluster the samples, based on an arithmetic mean and the classification depending on the pre-existing similarity among the samples investigated or added to the analysis ». Therefore the individual research of similarity approach (Knapp et al., 2017) is also relevant because distance calculation between two samples is a stable value.

­     6. Is it possible to perform partitioning cluster analyse/ classification/learning  experiments to strengthen the data regarding independent categories of the profiles with a linear support vector machine?

With the advent of new technologies associated to A.I., it could be certainly possible in the futur to improve the EmsB analysis by machine learning.

  1. Is there data available regarding the PNM, kind of lesion in ultrasound of the lesions?

We planned to compare the PNM classification and the genetic diversity of E. multilocularis in human but many cases are prior to the establishment of this classification mainly based on CT-scan imaging, or medical data were not available. The number of patients with available data was not large enough to propose a comparison, and a retrospective assessment of all lacking patients’ data files is not feasible anymore, and a respective patients’ consent can also not anymore be generated unfortunately. We add this information to the manuscript, lines 327-331: “The PNM system permits the clinical classification of alveolar echinococcosis [35,36] and it could be relevant to compare to EmsB profiles. Unfortunately, we only had available in the present study on third of the PNM data for the studied patients. Because of the lack of data we decided to deal with this subject in a future study”.

  1. Is the presence of the marker analyzed completely excluded in humans?

The specificity of the EmsB target was tested in Bart et al. 2006 on rodent and human DNA. Moreover, amplification was performed with pure E. multilocularis DNA and mixed E. multilocularis and rodent or human DNA and did not influence the fragment size analysis result.

  1. The lesion in humans is more or less completely necrotic. Have there been done experiments on necrotic metacestodic material to exclude artifacts due to high fragmentation of DNA in the necrosis? Is it possible to perform 4bp- Repeats rather than 2bp-tepeats in order to have a higher accuracy?

In the present study, no experiments were performed on necrotic material to test the consequences of DNA fragmentation on the EmsB amplification (i.e. “abortive” cases were not included, only active AE cases). Nevertheless, FFPE samples potentially sensitive to DNA fragmentation were tested in the present study and for one patient we had available frozen and FFPE isolates. Workable EmsB results were then obtained from FFPE.

  1. Please show examples of all the 9 profiles for comparison.

The 9 profiles were already presented on the Figure 3 but we add information on the manuscript to emphasize this data and we improved the figure for a better presentation of the electrophoregrams as example of each profile (line 204)), Lines 186-187: « Sixty-two patients clustered into nine profile groups, named P1 to P9 and presented on the Figure 3 (…) ».

Reviewer 2 Report

Ms presents interesting results on E. multilocularis microsatellite diversity among AE patients from four countries. The authors compared profiles of microsatellites between patients treated in long-term period and with different location of metacestodes (hepatic and others). Additionally, the authors put a lot of effort in validation of their methods, especially regarding the use of different labs, reagents and equipment. Its valuable contribution to our knowledge on alveolar echinococcosis.

Minor comments

Figure 6: in my version this fig is of poor quality, blurred. Is the legend on fig duplicated?

Discussion: there are several parts, especially regarding comparison of present Pn profiles with previously recognized ‘assemblages’ and profiles (Gn), not clear enough.

Lines 267-268: not clear, please clarify/expand

Lines 2790280: not clear, please clarify/expand; no info on present classification of Pn profiles to assemblages is provided in results.

Line 284: ? not clear;  how was similarity of profiles P4, P5 to G21 assessed?

Lines 289-291:

'Search for individual similarity’ how defined? Visual inspection? No info in results

Conclusions: the authors may consider adding short conclusion subchapter.

Author Response

RESPONSE TO THE REVIEWER 2

Comments and Suggestions for Authors

Ms presents interesting results on E. multilocularis microsatellite diversity among AE patients from four countries. The authors compared profiles of microsatellites between patients treated in long-term period and with different location of metacestodes (hepatic and others). Additionally, the authors put a lot of effort in validation of their methods, especially regarding the use of different labs, reagents and equipment. Its valuable contribution to our knowledge on alveolar echinococcosis.

Minor comments

  1. Figure 6: in my version this fig is of poor quality, blurred. Is the legend on fig duplicated?

As regards to the referee comment on the figure 6, we provide in the present manuscript a figure with better quality and clarification in the figure legend (Line 244). We merge the figures 6 and 7 for a better presentation of the data.

  1. Discussion: there are several parts, especially regarding comparison of present Pn profiles with previously recognized ‘assemblages’ and profiles (Gn), not clear enough.

As suggested by the referee we clarified and simplified this part in the manuscript (lines 347-357).

  1. Lines 267-268: not clear, please clarify/expand

As suggested by the referee, we clarified the paragraph. Lines 275-277: « As highlighted by the individual research of similarity and mapping (Figure 6b, Table S4) [22,31], Some some of these profiles had already been described locally in animals (profiles P5, P6 and P9), but others, as locally clustered among patients., Other profiles seemed to be widespread in foxes in Europe (profiles P2, P3, P7). ».

  1. Lines 279-280: not clear, please clarify/expand; no info on present classification of Pn profiles to assemblages is provided in results.

As suggested by the referee, clarifications were made on this point. Lines 292-295: « In our previous studies, based on the global shape of the different EmsB electrophoregram (number and position of the peaks) shapes were clustered into ,“assemblages” were described [29], in which diversity was associated with different profiles».

  1. Line 284: ? not clear;  how was similarity of profiles P4, P5 to G21 assessed?

The comparison between present profiles P4, P5 with the previously described profile G21 was assessed by an individual research of similarity and mapping, based on genetic similarity calculation and data from the previous studies by Knapp et co-workers, 2009, 2017. We clarified this point here. Lines 274-276: « As highlighted by the individual research of similarity and mapping (Figure 6b, Table S4) [22,31], some of these profiles had already been described locally in animals».

  1. Lines 289-291: 'Search for individual similarity’ how defined? Visual inspection? No info in results

The individual research of similarity is based on a comparison of the genetic distance between the tested sample with the whole EWET collection of samples, with a sorted list based on the distance matrix. We clarified this point in the present manuscript. Lines 310-312: « This result highlights the importance of using two classification approaches (dendrogram based on hierarchical clustering analysis and searches for individual similarity based on sorted lists from distance matrix) ».  

  1. Conclusions: the authors may consider adding short conclusion subchapter.

As suggested by the referee, the authors added a short conclusion to the manuscript, lines 349-358: « The genetic diversity of E. multilocularis parasite isolates from European human AE patients was assessed for the first time with the highly polymorphic EmsB microsatellite marker. This genetic diversity was compared to the EWET collection of reference, mostly composed of parasite specimens from foxes. Patients living in a highly endemic area presented common EmsB profiles. These profiles were described in foxes in limited geographical area for some of them or largely in Europe for others. Moreover, some EmsB profiles were described among patients over a period of 30 years. Thanks to this study, patients and animals were described to basically share the same EmsB profiles in Europe. Even if considered as an aberrant and dead end host, this present work allowed us to document the indirect involvement or position of humans within the E. multilocularis parasite life cycle. As regards to these findings, one can speculate no intermediate host selection is achieved by the parasite strains ».

Round 2

Reviewer 1 Report

No further comments. Nice work, will spur further analysis.